# Extended topological valley-locked surface acoustic waves

Ji-Qian Wang[1,2], Zi-Dong Zhang[1], Si-Yuan Yu [1,2,3✉], Hao Ge[1], Kang-Fu Liu[4], Tao Wu [4], Xiao-Chen Sun[1], Le Liu[1,2], Hua-Yang Chen[1], Cheng He [1,2,3], Ming-Hui Lu [1,2,3✉] & Yan-Feng Chen [1,2,3✉]

Stable and efficient guided waves are essential for information transmission and processing. Recently, topological valley-contrasting materials in condensed matter systems have been revealed as promising infrastructures for guiding classical waves, for they can provide broadband, non-dispersive and reflection-free electromagnetic/mechanical wave transport with a high degree of freedom. In this work, by designing and manufacturing miniaturized phononic crystals on a semi-infinite substrate, we experimentally realized a valley-locked edge transport for surface acoustic waves (SAWs). Critically, original one-dimensional edge transports could be extended to quasi-two-dimensional ones by doping SAW Dirac "semi-metal" layers at the boundaries. We demonstrate that SAWs in the extended topological valley-locked edges are robust against bending and wavelength-scaled defects. Also, this mechanism is configurable and robust depending on the doping, offering various on-chip acoustic manipulation, e.g., SAW routing, focusing, splitting, and converging, all flexible and high-flow. This work may promote future hybrid phononic circuits for acoustic information processing, sensing, and manipulation.

[1] National Laboratory of Solid State Microstructures & College of Engineering and Applied Sciences, Nanjing University, Nanjing 210093, China. [2] Collaborative Innovation Center of Advanced Microstructures, Nanjing University, Nanjing 210093, China. [3] Jiangsu Key Laboratory of Artificial Functional Materials, Nanjing University, Nanjing 210093, China. [4] School of Information Science and Technology, ShanghaiTech University, Shanghai 201210, China. ✉email: yusiyuan@nju.edu.cn; luminghui@nju.edu.cn; yfchen@nju.edu.cn

Surface acoustic waves (SAWs) have gradually become an essential part of modern civilization since interdigital transducers (IDTs) were invented in 1965[1]. Using IDTs to generate, receive and control on-chip SAWs in wavelengths ranging from $10^{-8}$ to $10^{-4}$ m, the unique advantages of SAWs can be well applied to today's microwave electronics in wireless communicating, sensing, navigating[2–5]. Compared with electromagnetic waves, SAWs of the same frequency have a much shorter wavelength (up to $10^{-5}$), which dramatically increases the possibility of device miniaturization. Also, as quasi-two-dimensional waves on solids, the transmission mode of SAWs is relatively simple, easy to distinguish, and low loss. Today, SAW technology is rapidly developing in the interdisciplinary fields of photonics, biomedical, and quantum science, giving birth to a significant number of interests in, e.g., acousto-optic modulations[6–8], phonon-based quantum information[9–13], spin–phonon interactions[14,15], and acoustic microfluidics[16,17].

All of the above studies are expected to obtain more precise SAWs manipulation, making high-quality SAW waveguide a key research front. Recently, several kinds of chip-scale SAW waveguides have been put forward, such as those based on guiding modes of phononic crystals[18,19], suspended beam structures[20,21], and ridges structures obeying impedance mismatch principle[14,22]. These cases successfully demonstrated the effective operation of SAWs accompanied by rich acoustoelectric and acousto-optic coupling effects. However, there are two fundamental challenges in this direction, i.e., (1) how to eliminate the backscattering that readily occurs when the SAWs detour quickly or passes through intersections, (2) how to match efficiently broadband SAW waveguides with broadband IDTs without causing more insertion loss and reducing the signal-to-noise ratio of the whole electro-acoustic devices. With the rise of research on topological effects in condensed matters, analogous quantum Hall families[23–30] have currently been adapted to bosonic systems, e.g., photonics[31–47], mechanics[48–61], airborne sounds[62–69], and even SAWs[70]. The core of these explorations is the gapless edge states, holding one-way transport with strong immunity to bendings and various defects.

In this article, we further use the topological advantage to propose and experimentally realize a monolithic SAW technology to simultaneously solve the aforementioned essential challenges. In principle, by designing the band structure of the SAWs in artificial periodic microstructure on a semi-infinite substrate, we have realized "Dirac semimetals" and "insulators" for the SAWs, respectively. Then, inspired by a band engineering methodology recently proposed in sonic[65] and electromagnetic[66] crystals, we spliced our SAW "semimetals" and "insulators" and successfully constructed a new kind of SAW guiding states with simultaneously topological protection and a wide equivalent width, called SAW extended topological valley-locked states (ETVSs). Through experiments, we have verified that these ETVSs have anti-reflection ability, high-flow, considerable working bandwidth for SAWs, and are configurable, demonstrating promising prospects for future large-scale phononic integrated circuits with versatile applications.

## Results

### SAW valley vortices and valley-locked states on a piezoelectric LiNbO₃ half-space.
The Quantum Valley Hall Effect (QVHE) was first found in two-dimensional (2D) hexagonal crystals, e.g., graphene[71], double-layer graphene[72], and transition metal dichalcogenides[73]. It originates as a result of the broken space inversion-symmetry. In those quantum states of matters, Dirac-fermions that correspond to different valleys move to opposite transverse edges in the presence of an in-plane electric field. In recent years, analogous to the Fermi electronic system, bosonic QVHE for photons[41–47] and phonons[54–60,65–69] have also been demonstrated. In this work, our first step is to realize the valley vortices[66,73] and the QVHE for SAWs. SAW phononic crystals are artificial mechanical microstructures based on semi-infinite substrates developed in this century, which can realize the dispersion modulation for SAWs, deriving, e.g., SAW bandgaps[74–79], guiding[18,19,80,81], localization[82]. By utilizing a pillar-type SAW phononic crystal[19,75,79], we have successfully mimicked the phenomenon of electrons in those 2D hexagonal crystals. Specifically, honeycomb-patterned mechanical micro-resonator pillars, coupled with each other through a half-infinite substrate, constitute our phononic crystal. To ensure electrical pumping/transducing for the SAWs, a piezoelectric crystal [in particular, a $y$-cut lithium niobate (LiNbO₃)] is chosen as our substrate. A schematic diagram of our phononic crystals is shown in Fig. 1a–c, there are two inequivalent phononic 'atoms' (micro-resonator pillars) in each unit cell, marked as pillar A and pillar B, respectively. These two sets of pillars have the same height ($h = 8$ μm) and side angle ($\theta = 6°$) but may differ in their radius.

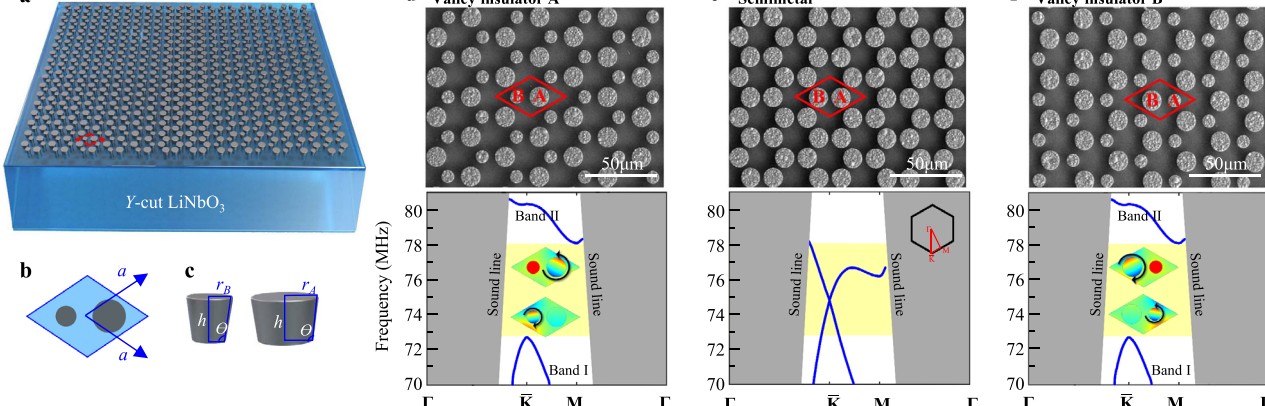

**Fig. 1 Our on-chip phononic crystals for SAWs. a** Schematic of our phononic crystals containing two sets of micro-resonator pillars, marked as A and B, on a $y$-cut LiNbO₃ semi-infinite half-space. **b** Unit cell of our phononic crystal, its lattice constant $a = 42/\sqrt{3}$ μm. **c** schematic of two inequitable pillars. **d–f** (upper) SEM images and (bottom) calculated band structures of three different phononic crystals with different topological phases. **e** A SAW Dirac "semimetal" with the same A and B pillars ($r_A = r_B = 5.8$ μm). **d** and **f** Two SAW valley "insulators" with inequivalent A/B pillars in their radius, both having omnidirectional bandgaps at their $\bar{K}$ and $\bar{K}'$ points of the 1st Brillouin zones, i.e., the valleys. We call the "insulator" in **d** ($r_A = 5.9$ μm $r_B = 3.9$ μm) as valley insulator A (VA) and the "insulator" in **f** ($r_A = 3.9$ μm $r_B = 5.9$ μm) as valley insulator B (VB). Insets show the SAW flux vortices observed at their valleys.

We define a relative mass $\Delta m$ for all these A and B pillars and $\Delta m = \frac{r_A - r_B}{r_A + r_B}$ for their contrast. When $\Delta m = 0$ (e.g., $r_A = r_B = 5.8$ μm in Fig. 1e), a "semimetal" for SAWs could be formed[80], the symmetry of the honeycomb crystal preserves a pair of SAW Dirac degeneracy around the $\bar{K}$ and $\bar{K}'$ points of the 1st Brillouin zone (considering the influence of the anisotropic LiNbO₃ substrate, $\bar{K}/\bar{K}'$ are used here to distinguish them from the high symmetry points K/K' of the traditional honeycomb lattice. See details in SI Note 7). When $\Delta m \neq 0$ (e.g., $r_A = 5.9$ μm $r_B = 3.9$ μm in Fig. 1d; and $r_A = 3.9$ μm $r_B = 5.9$ μm in Fig. 1f), the spatial inversion symmetry (along the $y$-axis) of the crystal is then broken, thus leading to "insulators" for SAWs with omnidirectional bandgaps.

In these SAW insulators, flux vortices of the SAWs could be observed at their frequency extrema $\bar{K}$ ($\bar{K}'$) points [insets of Fig. 1, and Fig.S9 of Supplementary Information (SI)], which is the characteristic of the classic wave analogous to the electronic valley states[73]. By constructing zigzag boundary between different SAW insulators in Fig. 1d and f, we realized the analogous QVHE for SAWs, or say SAW topological valley-locked states (TVSs). Simulation and experimental results are shown in Fig. S10 of SI. Note that the symmetry of the $y$-cut LiNbO₃ substrate slightly affects our phononic crystal's symmetry. However, this effect is faint and does not affect the presence of TVSs (see SI Note 7).

**Extended topological valley-locked states for SAWs.** Waveguides built from topological edge states may provide a disruptive advantage over conventionally designed ones, e.g., backscattering suppression, single-mode operation, and linear dispersion at their working bandwidth. It is especially the case for SAWs. Due to the relatively low impedance mismatch (generally <10¹) in different solid-state materials, and the extensively high sensitivity of acoustic waves in solid-state media, conventional designed solid-state acoustic waveguiding are pretty susceptible to intersections, sharp bends, and wavelength-scale defects.

Meanwhile, for (future) electro-acoustic SAW devices with SAW waveguides functioned between dual- (or multiple-) port IDTs, there is a crucial trade-off between the signal fidelity and the working bandwidth[17]. Broadband operating frequency is one of the core requirements of advanced signal processing. Generally, a broadband IDT requires a relatively large "aperture/wavelength" ratio, which is bound to reduce the proportion of the SAWs injected into the waveguide to the total SAWs that the IDT pumped. For the device as a whole, this requirement would increase the insertion loss of the waveguide signal and reduce its fidelity (see more in SI, Note 10). Consequently, to the practical use of SAW waveguides in concreting, e.g., phononic integrated circuits, a topological SAW waveguide with a wide aperture that can match broadband IDTs, is highly promising.

In this work, our second step (and the main achievement) is to realize such wide-aperture topological waveguides for SAWs by bringing about the ETVSs. Specifically, as shown in the upper panel of Fig. 2, based on the valley-type phononic crystal present in Section I, we doped our SAW semimetal ($\Delta m = 0$, $r_A = r_B = 5.8$ μm) in the interface between two SAW insulators with opposite valleys [i.e., one $\Delta m = 0.2$ ($r_A = 5.9$ μm $r_B = 3.9$ μm) and one $\Delta m = -0.2$ ($r_A = 3.9$ μm $r_B = 5.9$ μm)], thus directly making a heterostructured interface. This doping could extend the one-dimensional (1D) TVSs to quasi-two-dimensional (quasi-2D) ETVSs, with an additional degree of freedom (DOF) in the width of the doping area. Topological characteristics carried by the bulk on both sides of the heterostructured interface will tunnel through the doping area with Dirac dispersion[83,84], thus forming the ETVSs.

We numerically modeled several of those heterostructured interfaces with increasing doping areas. Projected band structures of these heterostructured interfaces and their SAW field/phase distributions are shown in the middle and bottom panels of Fig. 2. All dispersions of these ETVSs share the same characteristics as the conventional TVSs. Differently, for the TVSs, their energy is highly localized around the 1D interfaces. For the ETVSs, their energy fills the entire 2D doping area, thus offering a topological waveguide with a much wider aperture. As the doping area increases, the waveguide aperture may expand several times or even dozens of times to the original (none doped) one, at the price of a gradual decrease in working bandwidth. (see SI, Note 8).

Experimentally, we fabricated three different heterostructured samples with 3, 5, and 7 molecular doping layers. All the samples are in the same configuration, i.e., phononic crystal containing the heterostructured interface is prepared in the middle; two broadband IDTs, acting as SAW emitter and SAW receiver, are prepared on both sides, respectively. In the experiments, the emitter IDTs pump planar SAWs into the heterostructured interfaces (i.e., the waveguides), exciting the ETVSs. A laser vibrometer imaged the out-of-plane displacement of the excited ETVSs, confirming the SAWs do strongly localized in the whole doping area, together with uniform phases. After the SAWs passed through the heterostructured interfaces, they were captured by the receiver IDTs and read out by a network analyzer, demonstrating considerable transmittance (i.e., $S_{21}$, Fig. S18d, f, h of SI) at their operating frequencies. For the 3, 5, and 7-layer doping ETVSs, their maximum working bandwidths are expected to reach 7.8%, 6.7%, and 5.3%, respectively, without additional challenge for sample processing (see SI, Note 9).

**ETVS based splitter with high SAW throughput.** A three-port SAW splitter sample was fabricated to confirm that the ETVSs hold the same topological properties as TVSs. As shown in Fig. 3, this sample consists of five different parts, thus forming four ports labeled in the figure. Ports #1 and #3 are located on the straight SAW transmission route but are different in their valley pseudospins. Ports #2 and #4 are located on two "zigzag" SAW transmission routes with 120° sharp bends but hold the same valley pseudospin as Port#1. Planer SAWs wave generated by IDTs and are injected into the multi-port device through Port#1.

In conventional SAW waveguide devices, the input SAWs will mainly directly exit the device via Port #4 but partially go through the "zigzag" routes to reach Ports #2 and #3. However, in this topological valley-locked waveguide, the input SAWs from Port #1 could only exit the system via the ports that hold the same valley-pseudospin. Hence, these SAWs should pass through those "zigzag" routes and eventually be captured at Ports #2 and #3. Experimental imaged SAW distributions around the input and output ports confirm that the SAWs experience this "Port#1 → Ports# 2/3" transport. These results agree well with theoretical expectations, proving that in our SAW system, the ETVSs do hold the same function as conventional TVSs. Our SAW splitter is a demo for the valley-pseudospin locking and a promising primary component for future SAW integrated circuits with high throughput and energy capacity.

**Defects immunity and phase reconstruction for SAWs.** In the QHEs and QSHEs[35,85], the nonzero (spin) Chern number originates from integrating the Berry curvature over the whole Brillouin zone. For the QVHEs[58,66,69], although the integration is zero, the Berry curvature is highly localized around the K and K' points (i.e., the valleys), giving birth to nonzero valley Chern number with opposite signs at paired valleys.

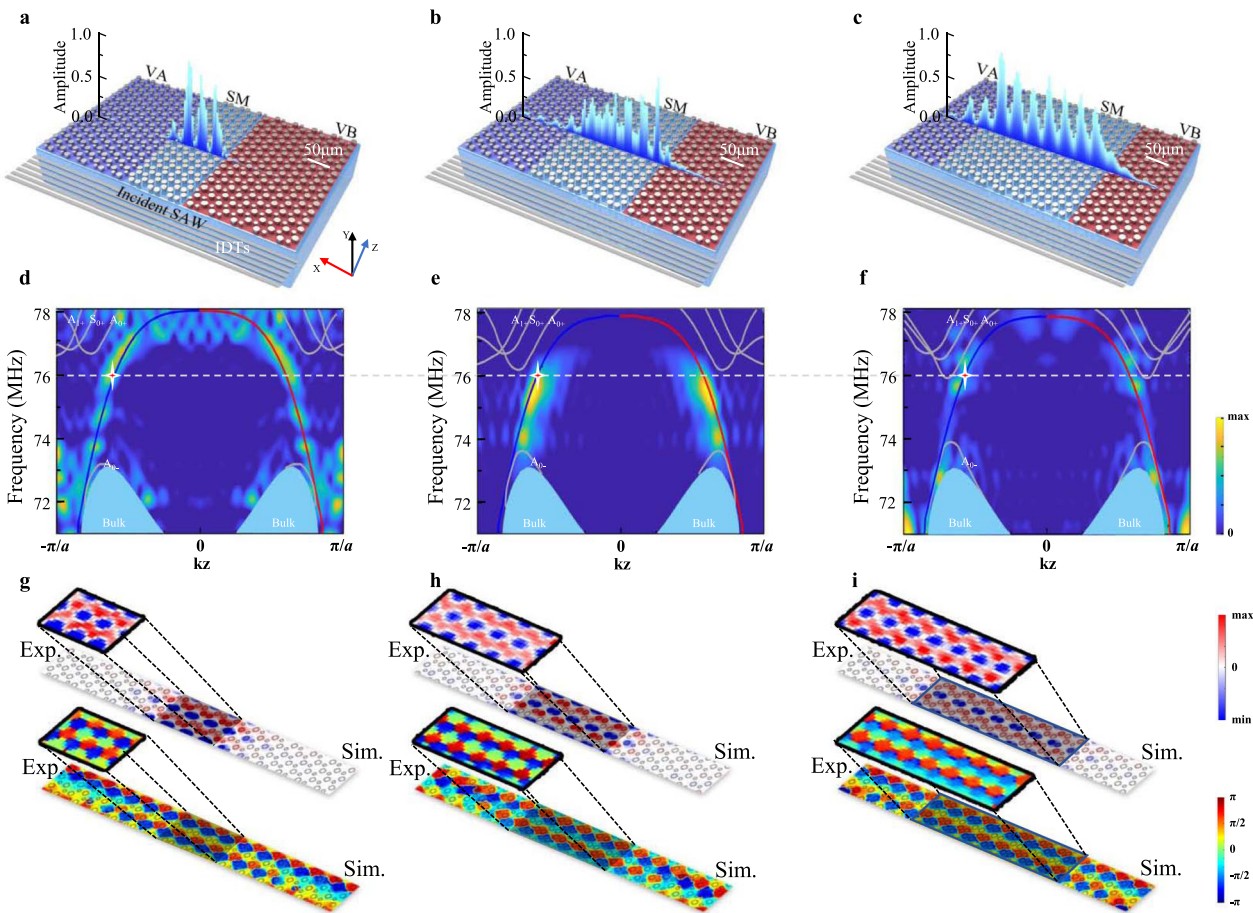

**Fig. 2 Extended topological valley-locked states for SAWs. a–c** (bottom) Schematics of three heterostructured valley interfaces (see SEM images in Fig. S2), with increasing molecular layers (from 3, 5, to 7, respectively) of SAW "semimetal" (SM) doped inside a VA and a VB. (upper) Experimentally measured SAW out-of-plane amplitude distributions crossing the doping areas. **d–f** Calculated and experimentally measured band structures of the heterostructured interfaces. The color bar represents the density of energy. **g–i** Simulated and experimentally imaged SAW out-of-plane displacement and phase distributions of the heterostructured interfaces (at 76 MHz shown in the figures), demonstrating a pseudo-diffusion-like phase uniformity of these edge states.

When a wavelength-scale defect is introduced into a conventional QVHE system, the Berry curvature will become less localized, and the intervalley mixing will become stronger, thus destroying the TVSs[61]. Such destruction can be observed in our SAW system, as shown in the left panel of Fig. 4. We experimentally prepared a conventional (non-doping) valley-locked edge with two wavelength-scale vacuum defects deliberately embedded. When time-harmonic SAWs are pumped into this edge, relatively few SAWs can be detected from the exit. Meanwhile, the phase of the transmitted SAWs is wholly disturbed, confirming the destruction of the edge states.

Unlike non-doped TVSs in conventional QVHE systems, ETVSs have an excellent tolerance for the same wavelength scale defects. The additional width (DOF) of ETVSs endows them with stronger robustness in their transmission routes since the same defect should impose much less disturbance on quasi-2D ETVSs than 1D TVSs. This robustness was observed in a comparative experiment, as shown in the right panel of Fig. 4. When time-harmonic SAWs are pumped into a doped edge, much more SAWs can now be detected from the exit. Along the transmission route, the phase of the SAWs is disturbed around the defect. However, the phase uniformity would quickly emerge as the SAWs propagate away from the defect for several wavelengths. This intriguing phenomenon would bring us some novel SAW applications with precise wavefront control, e.g., SAW cloaking, focusing, collimating, and beam expanding.

**ETVS's stability and configurability**. The existence of the topological edge states is originated and preserved by the non-trivial topology of the bulk bands. Although the characteristics and functionalities of those edge states, e.g., defects immunity and insensitivity of the external field perturbation, have been widely demonstrated in electronics, photonics, and phononics, doping (and manipulating) on these topological edges are currently underdeveloped and reveals a promising research area to explore complex nontrivial physics and applications[86] in, e.g., signal processing and information communications.

For our present SAW ETVSs, we will demonstrate their stability and configurability to the different doping. Our experiments are shown in Fig. 5. The doping width is adjustable by adopting different doping layers, i.e., layers of the SAW semimetal embedded in the doping area. As the doping width increase, the frequency range in which the ETVSs exist would continue to narrow and eventually disappear. Except that, in a fixed doping width, the doping "height" is also adjustable. This "height" can be defined as the Dirac frequency of the SAW semimetal. By changing the radius of the micropillars, the doping "height" changes from 77.90 MHz, 75.99 MHz, 74.94 MHz (Fig. 5a, b, and c, respectively). Calculated projected band structures of these different heterostructured interfaces (see Fig. S20 of SI) show that the ETVSs crossing the whole bulk band stably exist in all these cases, except for slight frequency shift.

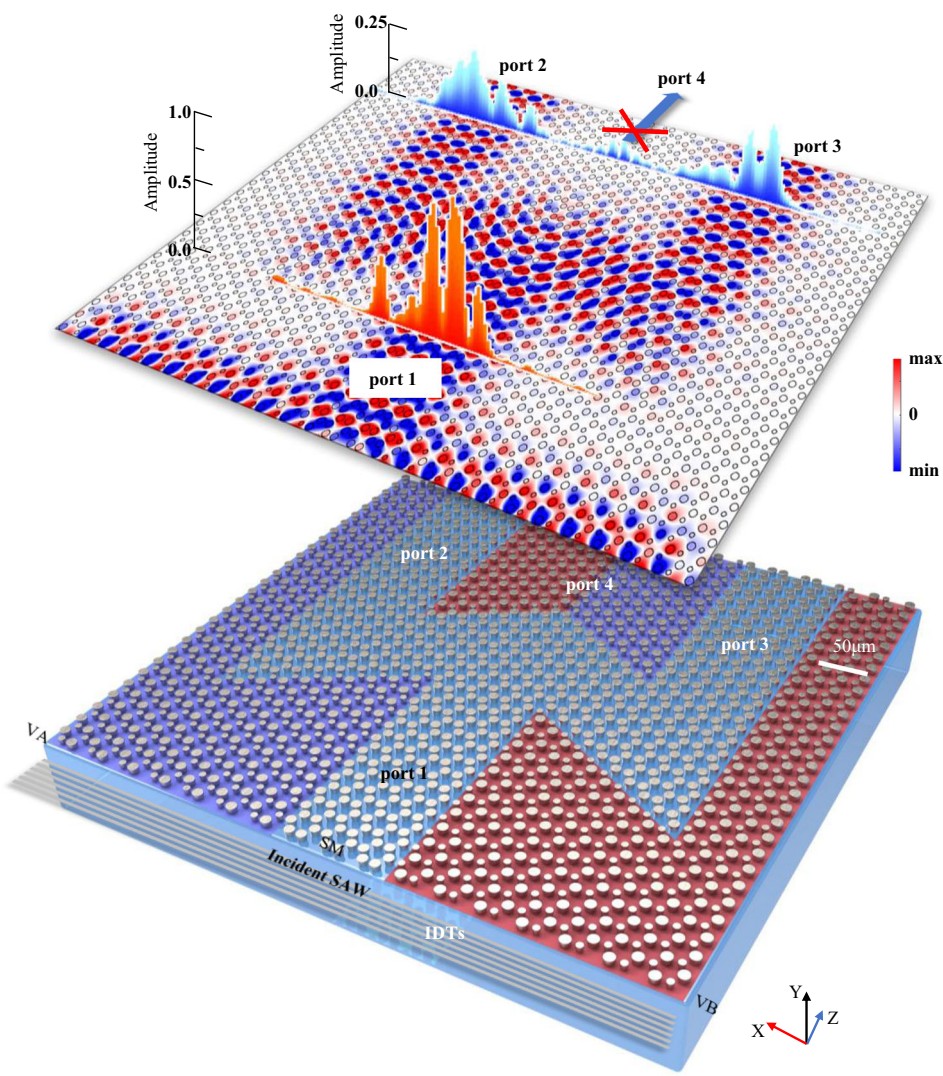

**Fig. 3 High throughput SAW splitter based on the ETVS.** (bottom) Schematic of a three-ports SAW splitter sample consists of the (blue shaded) VA, (red shaded) VB, and (unshaded) SM. In the experiment, planar SAWs are injected into the sample from Port#1 and captured from the other three ports. (upper) Calculated SAW out-of-plane displacement distributions and experimentally imaged SAW amplitude around all four ports. All SAWs are highly localized in routes that hold the same valley pseudospin, *i.e.*, the SAWs are output the sample only via Ports #2 and #3.

Experimentally, these configurable ETVSs are imaged (see Fig. 5e–g), all exhibit phase uniformity with slightly changed patterns. When the doping "height" is relatively high, the interaction between adjacent micropillars seems to be relatively weak; SAWs in the doping area (i.e., the ETVSs) are more similar to plane waves. As the doping "height" decreases, the inter-pillar coupling becomes stronger; SAWs in the doping area are more localized around the micropillars, exhibiting a "fish scales" style. Notably, the transmittances of the three ETVS waveguides are unity since they are all valley pseudospin locked and robust to wavelength-scale defects and interactions. Practically, it provides a new strategy to manipulate SAWs in the ETVS waveguide, e.g., operation frequencies and distributions[84].

## Discussions
We proposed and experimentally verified a SAW analog of quantum valley Hall system, accompanied by the ETVEs for SAWs, *i.e.*, an ideal candidate for SAW integrated circuits. The SAWs ETVEs are anti-reflection (making the waveguide have a high degree of freedom and low loss), high-flow and configurable. Based on them, a series of application-driven SAW prototype devices, e.g., waveguides and beam splitters with flexible path and defects immunity, are fabricated and experimentally demonstrated. Although the operating frequency of the SAW prototypes in this article is in the tens of megahertz, future devices based on the same principle can exceed several gigahertz or even higher through advanced acoustic MEMS manufacturing technology. Similarly, the material systems used to build this type of ETVSs can also be varied, such as traditional piezoelectric materials for SAWs (e.g., $LiTaO_3$, AlN, GaN), mainstream silicon-based materials (e.g., Si, $SiO_2$, SiN, SiC), and mechanical or even piezoelectric 2D materials (e.g., monolayer transition metal dichalcogenides, group III–V binary compounds and group IV monochalcogenides). Future designs can comprehensively consider using these materials based on their piezoelectric characteristics, electromechanical coupling characteristics, velocity, temperature drift, loss, and convenience of micro-nano processing. This universal design principle may flourish and promote future topological-related applications, such as (topological) photonic/phononic integrated circuits for classical and quantum photonics/phononics. One challenge is how to increase the working bandwidth of these ETVSs further. If the bandwidth can

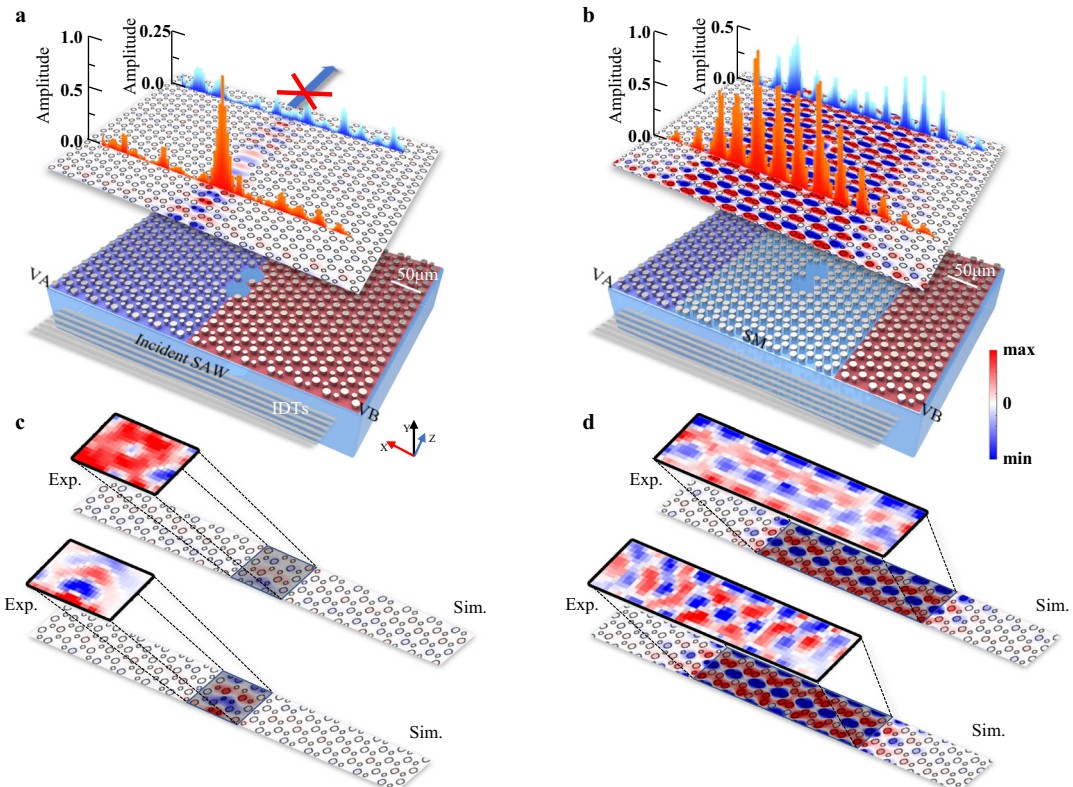

**Fig. 4 Comparison of wavelength-scale defect robustness between the TVS and ETVS for SAWs.** (Left panels) the TVS case; (right panels) the ETVS case. **a**, **b** (bottom) Schematics of a TVS waveguide and an ETVS waveguide have two uniform wavelength-scale vacuum defects embedded. Blue, red, light blue shaded areas are the VA, VB, and the defects; the SM remains unshaded. In the experiment, the same planar SAWs are injected into these two samples. (upper) Calculated SAW out-of-plane distributions on the surface of the two samples. 1D red and blue bars are experimental measured SAW out-of-plane amplitude before and behind the defects. **c**, **d** Calculated and experimental imaged SAW out-of-plane displacement distributions before and behind the defects in both the waveguides.

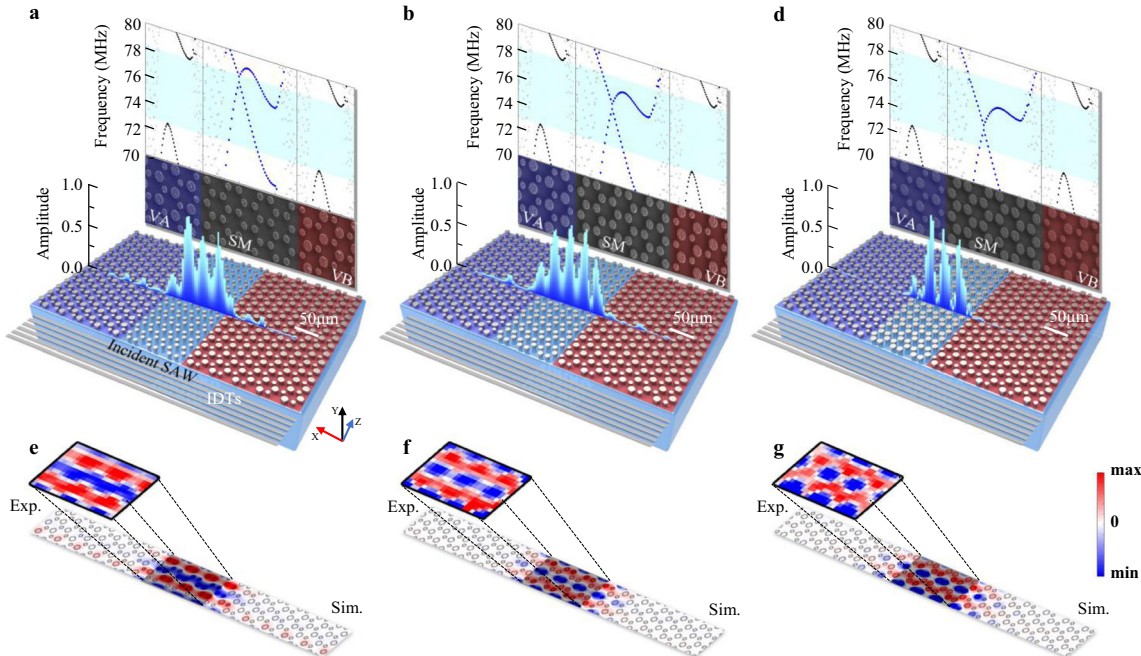

**Fig. 5 ETVS's stability to the doping of different energy barriers. a–c** (bottom) Schematics of three heterostructured valley interfaces, when the radius of the micropillars of SMs increasing from **a** 4.3 μm to **b** 5 μm to **c** 5.8 μm. (upper) SAW band structures of the corresponding SMs. Their SAW Dirac frequency decreases inside the VA/VB's bulk bandgap. (middle) Experimentally measured SAW out-of-plane amplitude distributions crossing the doping areas. SAWs remain highly localized in all these cases. **d–f** Experiment and simulation SAW out-of-plane displacement distribution (at 76 MHz shown in the figures) depending on different doping.

exceed 15% or even 25% (i.e., the largest bandwidth in current 5G mobile networks), it will significantly facilitate their demonstrating functionalities in practical scenarios. Some valuable results have recently appeared in broadband topology materials, e.g., those using topology optimization[87] and inverse design[88]. From the perspective of condensed matter physics, due to the easy-to-process characteristics of artificial structures and SAWs' intuitive and high-fidelity characteristics, this research on doped topological edge states may provide an ideal platform for exploring topological properties and transport behavior. It may inspire other related studies on bosons and fermions.

## Methods

**Sample fabrications**. We used the LIGA-like technique to fabricate our phononic crystals on the LiNbO$_3$ semi-half-space. It contains four following steps: a 10 nm Cr followed by a 50 nm Cu layer, working as a seed layer, is deposited on a 500 μm thickness y-cut LiNbO$_3$ substrate; (2) patterning of phononic crystal structures using UV lithography with 10 μm positive resist AZ9600; (3) Electrochemical plating of nickel (Ni) micro-resonator pillars on the exposed Cu-Cr seed layer; (4) removal of the resist.

**Numerical calculations**. Commercial finite element software COMSOL Multiphysics performs all full-wave simulations. Three-dimensional unit cells used in our band structure calculation are in triangular lattice, and contain two Ni pillars on y-cut LiNbO$_3$ semi-half-space. Floquet periodic boundary conditions are applied to the unit cells. For all samples in Figs. 2–5, low reflection boundaries are set along the $z$-direction, and the model's bottom and continuous boundaries are placed along the $x$-direction. The elastic parameters of the Ni pillars are density $\rho_{Ni} = 8906 \, kg \, m^{-3}$, Young's modulus $E_{Ni} = 175 \times 10^9 \, Pa$, and Poisson's ratio 0.30. Note that these values are specific to our growth technique.

**Reporting summary**. Further information on research design is available in the Nature Research Reporting Summary linked to this article.

## Data availability

All simulation data in the current study are performed using commercial finite element software COMSOL Multiphysics. All experimental measured SAW displacement/energy data are characterized by a commercial scanning vibrometer (Polytec UHF-120) with software PSV Acquisition. They are under private user license which cannot be made public. All data are available from the corresponding authors upon reasonable request.

## Code availability

Numerical simulations in this work are all performed using the commercial finite element software COMSOL Multiphysics. All related codes can be built with the instructions in the "Methods" section and available from the corresponding authors upon reasonable request.

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

## Acknowledgements

The work was jointly supported by the National Key R&D Program of China (Grant Nos. 2021YFB3801801, 2017YFA0305100, and 2017YFA0303702) and the National Natural Science Foundation of China (Grant Nos. 11890702, 92163133, 51732006, 52022038, and 61874073). We also acknowledge the support of the Natural Science Foundation of Jiangsu Province, Natural Science Foundation of Shanghai (Grant No. 19ZR1477000), and Fundamental Research Funds for Central Universities.

## Author contributions

S-Y.Y., M-H.L., and Y-F.C. conceived the original idea and supervised this project. J-Q.W. performed the numerical simulations. J-Q.W., Z-D.Z., and K-F.L. carried out the experiments. J-Q.W., S-Y.Y., H.G., X-C.S., L.L., T.W., and C.H. analyzed the data. J-Q.W. and S-Y.Y. wrote the manuscript with the assistance of H-Y.C., X-C.S., C.H., and T.W., All authors contributed to scientific discussions of the manuscript.

## Competing interests

The authors declare no competing interests.
