## [Peer review file · Nature Communications]

REVIEWER COMMENTS

Reviewer #1 (Remarks to the Author):

The manuscript "Extended topological valley-locked surface acoustic waves" by Ji-Qian Wang et al. presents the topological valley-locked waveguide states for surface acoustic waves, as an extension of the similar valley-locked waveguide states found for bulk acoustic waves. These surface waveguide states are shown to be robust against the defects and the doping. This topic is timely and interesting, and the devices, working at tens of megahertz frequency, have potential in applications. I would like to recommend it for publication in Nature Communications, but I have one question below for the authors to address.

In Fig 2d, it can be seen that there is an apparent discrepancy between the simulated and measured data regarding the band structure of the waveguides states. The experiment fails to capture the waveguide states in a wide frequency region inside the bandgap, while instead some unknown features appear in this region. Since the case of Fig.2d has the widest bandgap, one expects the best experiment results for the case among all the three cases of Fig.2d,e,f, but it seems it is not the case. Can the experiment in Fig2d be improved?

Reviewer #2 (Remarks to the Author):

In the submitted manuscript Wang et al developed a new surface acoustic wave (SAW) prototype device based on the extended topological valley-locked states (ETVSs). The authors show that the developed device exhibit valley pseudospin-momentum locking, and is immune to local defects. In my opinion, the idea of ETVSs is novel and the results present in this paper could attract interest in topology-related device applications. I would like to recommend the publication in Nature Communication after the following comments are addressed:

1. As the author mentioned in the manuscript the working bandwidth of the ETVSs eventually vanishes as the width of the semimetallic region (i.e. doping width as is called in the manuscript) increases. What is the critical width that ETVSs still exist?
2. The topological valley states are protected by valley Chern numbers which are well-defined only when the bandgap opened by the A-B sublattice asymmetry is not too large. This places a fundamental limit for the maximum working bandwidth of the ETVSs. What is this maximum working bandwidth?

Reviewer #3 (Remarks to the Author):

The paper reports on an experimental study of extended valley-locked surface acoustic waves (SAW) in a phononic crystal of pillar on a semi-infinite substrate of lithium niobate (LNO).

On the formal side, I don't find the paper well written and the language precise enough. The introduction isn't very convincing regarding the precise novelty and the literature on SAW phononic crystal is not well acknowledged. The qualities attributed to the proposed topological PC waveguide are too exaggerated compared to what has effectively been achieved (see subsequent comments). There is also a general lack of technical details and lack of physical discussion. For all all these reasons, I don't recommend publication in the present form and I question the novelty is sufficient for Nature Communications.

1. The concept of 'extended valley-locked' waves in topological crystals was proposed and discussed in ref. [65]; this fact is not clearly acknowledged in the paper.

Furthermore, the authors have recently published the paper [Zhang, Zi-Dong, Si-Yuan Yu, Hao Ge, Ji-Qian Wang, Hong-Fei Wang, Kang-Fu Liu, Tao Wu, Cheng He, Ming-Hui Lu, and Yan-Feng Chen.

"Topological Surface Acoustic Waves." Physical Review Applied 16, no. 4 (2021): 044008.] that is not even cited. To this reviewer, the two papers look very much alike, with the same C6v crystal of pillars selected as the starting design, though the authors might say that the topological parameter that is tuned is not the same (pillar diameter here, pillar position inside the unit cell there). The paper must be cited and the differences must be discussed.

2. How can the C6v crystal in the Physical Review Applied paper cited above show a double Dirac cone at the Gamma point, whereas figure 1 in the present paper shows a single Dirac cone at the K point? As a note, the working frequency is 76 MHz in both cases and the samples look exactly the same. Moreover, the complete band structure must be shown, not only the small frequency range around 76 MHz.

3. Is the anisotropy of LNO taken into account in the design and the numerical simulations? Does the C6v symmetry of the 2D phononic crystal survive the trigonal crystal system of LNO?

4. It is argued that the topological waveguides created are broadband. This statement is particularly misleading. The fractional bandwidth varies between between 1/15 and 1/25 (3 to 5 MHz around 75 MHz); I would classify such values as small bandwidth and this is a general problem for topological waveguides designed from the opening of the band gap that was originally closed by symmetry.

5. The figures need to be prepared with more care. The wavevector axis of phononic band structures shows inconsistent information. The SEM images are heavily processed (look like CAD images) and no details of the actual fabricated samples are provided. Color bars have no units or scale. The 4-port splitter in figure 3 is a 3-port splitter per design. The transmission values in figure S4 show strong insertion loss, from 11 dB to 26 dB; the conclusion that 'SAWs transmission [...] has unprecedented high throughput' has no ground and no comparison with previous results in the literature is presented.

Author Response to Reviewer Comments

Reviewer #1 (Remarks to the Author):

Reviewer Comment #1-1: The manuscript “Extended topological valley-locked surface acoustic waves” by Ji-Qian Wang et al. presents the topological valley-locked waveguide states for surface acoustic waves, as an extension of the similar valley-locked waveguide states found for bulk acoustic waves. These surface waveguide states are shown to be robust against the defects and the doping. This topic is timely and interesting, and the devices, working at tens of megahertz frequency, have potential in applications. I would like to recommend it for publication in Nature Communications, but I have one question below for the authors to address.

Our Response: We thank the reviewer for highlighting our effort and scholarly presentation of the work, as well as the explicit recommendation of publication in Nature Communications.

1. In Fig 2d, it can be seen that there is an apparent discrepancy between the simulated and measured data regarding the band structure of the waveguide states. The experiment fails to capture the waveguide states in a wide frequency region inside the bandgap, while instead some unknown features appear in this region. Since the case of Fig.2d has the widest bandgap, one expects the best experiment results for the case among all the three cases of Fig.2d,e,f, but it seems it is not the case. Can the experiment in Fig2d be improved?

Our Response: We appreciate the reviewer’s comment on the very important point around our experimental results demonstrated in this paper. In response, we re-run the experiment (including sample processing and testing) and put the new and improved experimental results in Fig.2d.

Revised Fig. 2 in the main text

Reviewer #2 (Remarks to the Author):

Reviewer Comment #2-1: In the submitted manuscript Wang et al developed a new surface acoustic wave (SAW) prototype device based on the extended topological valley-locked states (ETVSs). The authors show that the developed device exhibit valley pseudospin-momentum locking, and is immune to local defects. In my opinion, the idea of ETVSs is novel and the results present in this paper could attract interest in topology-related device applications. I would like to recommend the publication in Nature Communication after the following comments are addressed:

Our Response: We are grateful to the reviewer for highlighting the academic value of our work and supporting its publication in Nature Communications.

1. As the author mentioned in the manuscript the working bandwidth of the ETVSs eventually vanishes as the width of the semi-metallic region (i.e. doping width as is called in the manuscript) increases. What is the critical width that ETVSs still exist?

Our Response: We thank the reviewer for raising this important question. To better determine this value, we newly calculated the band structures of the ETVSs when the semi-metallic region increases from 15 to 39 layers, as shown in Fig. R2-1.

Figure R2-1 | a to j: band structures of the ETVs from 3 to 39 doping layers, with a step of 4.
k and j: SAW out-of-plane displacement distribution of modes A_{0+} and S_{0+} , respectively.

In R2-1a to R2-1j, the ice-blue areas indicate the bulk bands, and the red/blue solid line is the dispersion of ETVs. Solid gray lines represent the modes in the doping area. They can be further divided into symmetric (S) and antisymmetric (A) modes, as shown in R2-1k and R2-1l.

By measuring the bandgap of the A_0 modes, *i.e.*, the top of the A_{0+} modes and the bottom of the A_{0-} modes, the passing frequency for the ETVs can be determined. We plot these values with the number of doping layers in Fig. R2-2a. It can be seen that this bandgap narrows with the doping width increasing.

Figure R2-2 | **a**: (red dots) simulated bandgap of A_0 modes from 3 to 39 doping layers, and (black dotted line) corresponding approximate fitting. The two insulators besides the semi-metallic region are set as 8 layers in the simulation. **b**: band structure of the semi-metal.

Even if the number of layers reaches 39, the passing frequency still has 0.454MHz. Limited by the capabilities of our computers, we can hardly calculate the situation with a larger number of layers. Therefore, we performed an approximate fit to this trend, as shown in Fig. R2-2b. The fitting results show that when the number of doping layers reaches over 72, the A_0 bandgap may eventually close. The fitted closing frequency is about 74.94MHz, just the Dirac frequency of the semi-metal. It is reasonable because the Dirac semi-metal will gradually dominate the whole structure with the doping width increasing, where the interaction between two insulators becomes weaker and finally negligible. However, it should be noted that the exact value of the critical width of the semi-metallic region (*i.e.*, 72 layers in our simulation) highly depends on the boundary condition of the heterostructure (e.g., the width of the insulator besides the semi-metal). In the simulations of Figs. R2-1 and R2-2, both insulators besides the semi-metallic region are set as 8 layers. Moreover, with the change of the material and geometric parameters of the phononic crystal, this critical width may also vary to a certain extent.

Action Taken: According to the reviewer’s question, we have added a related description in the main text of our revised manuscript, as “As the doping area increases, the waveguide aperture may expand several times or even dozens of times to the original (none doped) one, at the price of a gradual decrease in operating bandwidth.” in the 4th paragraph of Section II. Also, we put the response to this question (including the figures) as Note 8 in the Supplementary Information (SI) of our article.

2. The topological valley states are protected by valley Chern numbers which are well-defined only when the bandgap opened by the A-B sublattice asymmetry is not too large. This places a fundamental limit for the maximum working bandwidth of the ETVs. What is this maximum working bandwidth?

Our Response: We appreciate the reviewer’s interest in the very important point about the maximum working

bandwidth of the ETVSs. We fully agree that the topological valley Chern numbers are well-defined only when the bandgap opened by the A-B sublattice asymmetry is not too large. The nonzero valley Chern numbers are based on highly localized nonzero Berry curvature at the K (\bar{K}) point. With the increase of A-B sublattice asymmetry, the distribution of Berry curvature at different valleys (K and \bar{K}) will gradually become smoother and overlapping to each other, thus invalidating the non-trivial topological characteristics of the system.

In our SAW system, the maximum effective working bandwidth of the ETVS is determined by two factors:

- (I) the bulk bandgap opened by the A-B sublattice asymmetry
- (II) the bandgap between S_0^- and S_0^+ . The antisymmetric (A) modes are not taken into consideration because they cannot be excited by the interdigital transducers (IDTs).

Hence, the overlapping bandwidth of the I and II determines the working bandwidth of the ETVSs. To determine the maximum bandwidth of I, we calculated the distribution of Berry curvature at the \bar{K} and \bar{K}' points of our phononic crystal when the A-B sublattice asymmetry gradually became stronger, as shown in Fig. R2-3.

Figure R2-3 | a-e: A-B sublattice asymmetry gradually increasing. f-j: corresponding band structures. k-o: corresponding Berry curvature distributions at the \bar{K} and \bar{K}' points.

Clearly, with the increase of A-B sublattice asymmetry, the intensity and localization of Berry curvature near the \bar{K} and \bar{K}' points gradually decrease. In Fig. R2-3e/j/o, the maximum/minimum of Berry is only 1/60 of that in Fig. R2-3a/f/k. Although the bandwidth at this time has exceeded 10MHz (~13%), the system's non-

trivial topological characteristics have become quite weak.

To find the maximum bandwidth of **II**, we calculated the projected band structures with different doping layers and increasing sublattice asymmetry, as shown in Fig. R2-4. Note that S_{0-} modes in Figs. R2-4 a/b/d/e/g are merged in the bulk bands.

Figure R2-4 | band structures for the EVTSs with different A-B sublattice asymmetry and doping layers. From left to right: increasing A-B sublattice asymmetry. **a\|d\|g**: $r_A = 5.9\mu\text{m}$, $r_B = 3.9\mu\text{m}$; **b\|e\|f**: $r_A = 6.3\mu\text{m}$, $r_B = 3.5\mu\text{m}$; **c\|f\|i**: $r_A = 6.7\mu\text{m}$, $r_B = 3.1\mu\text{m}$. From up to bottom: increasing doping layers, *i.e.*, 3, 5, 7 layers, respectively.

When the doping layers are constant, the S_0 bandgap is almost fixed; this value does not change with A-B sublattice asymmetry. This is understandable because the S_0 mode exists in the doping area (*i.e.*, the semi-metallic region), so it is hardly affected by the two insulators with the A-B sublattice asymmetry. Notably, when the A-B sublattice asymmetry is increased to a certain extent, the overlapping bandwidth of the **I** and **II** bandgaps is only determined by **II**. Therefore, in our SAW phononic crystals, for 3, 5, and 7-layer doping ETVSSs, their maximum working bandwidths are 5.74MHz (7.8%), 5MHz (6.3%), and 3.97MHz (5.7%), respectively.

Action Taken: According to the reviewer's question, we have added a related description in the main text of

our revised manuscript, as “For the 3, 5, and 7-layer doping ETVs, their maximum working bandwidths are expected to reach 7.8%, 6.7%, and 5.3%, respectively, without additional challenge for sample processing.” in the 5th paragraph of Section II. Also, we put the response to this question (including the figures) as Note 9 in the SI.

Reviewer #3 (Remarks to the Author):

Reviewer Comment #3-1: The paper reports on an experimental study of extended valley-locked surface acoustic waves (SAW) in a phononic crystal of pillar on a semi-infinite substrate of lithium niobate (LNO). On the formal side, I don't find the paper well written and the language precise enough. The introduction isn't very convincing regarding the precise novelty and the literature on SAW phononic crystal is not well acknowledged. The qualities attributed to the proposed topological PC waveguide are too exaggerated compared to what has effectively been achieved (see subsequent comments). There is also a general lack of technical details and lack of physical discussion. For all these reasons, I don't recommend publication in the present form and I question the novelty is sufficient for Nature Communications.

Our Response & Action Taken: We appreciate the reviewer for his/her careful review and comments, although we regret the reviewer's evaluation of our work. We carefully consider that the reviewer may misunderstand our current work's physical background and its key value for applying "topological waveguides" to practice. We will elaborate on the specifics below, and we sincerely hope to get the reviewer's understanding and approval.

As per the reviewer's comments above,

- 1) We have carefully reorganized our language to make this paper more detailed and accurate.
- 2) Regarding the literature on SAW phononic crystals, we have revised our manuscript as "SAW Phononic crystals are artificial mechanical microstructures based on semi-infinite substrates developed in this century, which can realize the dispersion modulation for SAWs, deriving, e.g., SAW bandgaps⁶⁹⁻⁷⁴, guiding^{18,19,75,76}, localization⁷⁷. By utilizing a pillar-type SAW phononic crystal^{19,70,74}, we..." in the 1st paragraph of Section I. Please note, however, that the focus of our current work is not solely on the realization of the SAW phononic crystal, but on the physical effects and device functions that the material system shows. We appreciate the reviewer's understanding in this matter.
- 3) We have added more technical details, including material parameters for all our simulations, the preparation process of our phononic crystals, etc. See Notes 1 and 2 in our new Supplementary Information (SI).
- 4) We have added and highlighted more physical discussion, including the calculations for Berry curvatures, quantum valley Hall effect, etc. See Notes 4 to 7 and 9 in our new SI.

Below, we will respond to the reviewer's subsequent comments.

Reviewer Comment #3-2:

The concept of 'extended valley-locked' waves in topological crystals was proposed and discussed in ref. [65]; this fact is not clearly acknowledged in the paper.

Our Response & Action Taken: We appreciate the reviewer for his/her careful review and comments. The concept of 'extended valley-locked' waves was pioneered by Wang *et al.* in a sonic crystal system [*Nat. Commun.* **11**, 3000, (2020)] with similar implementation in a time-reversal symmetry broken electromagnetic system [*Phys. Rev. Lett.* **126**, 067401, (2021)]. In our original manuscript, these two papers were cited just after the introduction of SAWs and artificial topological crystals. According to the reviewer's comment, in our revised manuscript, we have revised the relevant descriptions to clarify the contributions of these papers and their inspiration to our work, as "Then, inspired by a band engineering methodology recently proposed in sonic⁶⁵

and electromagnetic⁶⁶ crystals...” in the 2nd paragraph of the introduction.

Here in response, we also want to clarify that our work is not only the realization of extended valley-locked states (ETVSs) for SAWs, but more fundamentally, the realization of valley-locked states (TVSSs) for SAWs. It provides a novel guiding SAW with simultaneous “anti-reflection” ability and high-flow, which will undoubtedly promote the application of topological physics for practical integrated acoustics.

Furthermore, the authors have recently published the paper [Zhang, Zi-Dong, Si-Yuan Yu, Hao Ge, Ji-Qian Wang, Hong-Fei Wang, Kang-Fu Liu, Tao Wu, Cheng He, Ming-Hui Lu, and Yan-Feng Chen. "Topological Surface Acoustic Waves." *Physical review Applied* 16, no. 4 (2021): 044008.] that is not even cited. To this reviewer, the two papers look very much alike, with the same C6v crystal of pillars selected as the starting design, though the authors might say that the topological parameter that is tuned is not the same (pillar diameter here, pillar position inside the unit cell there). The paper must be cited and the differences must be discussed.

Our Response & Action Taken: We appreciate the reviewer's attention in our previously published paper [*Phys. Rev. Applied.* 16, 044008 (2021)]. First of all, the current paper was submitted to Nature Communications on September 9th, 2021, while the *PRApplied* paper was accepted and published after that time and has not been submitted to any preprinted website (*e.g.*, arXiv) before. At the time of submission of the current paper, we could not foresee the *PRApplied* paper's publication information, so we could not cite it. *As per the reviewer's comment*, we have added a citation to the *PRApplied* paper in our revised manuscript.

Although these two papers seem somewhat similar to the reviewer (for example, both phononic crystals are made of pillars and operate at similar frequencies), they are very different in their related physics and application prospects. Now we discuss their differences in detail below.

I: Difference in physics

- The *PRApplied* paper is about analogous quantum spin Hall (QSH) states. The existence of these states was proposed in electronics in 2003¹⁻³ and be observed in materials, *e.g.*, HgTe⁴. In the absence of an external magnetic field (*i.e.*, under the time-reversal symmetry), there are conductive states on the surface of an insulator, and the spin of the surface state is locked at a right angle to the carrier momentum.

In recent years, QSH has been developed into classical wave systems. For example:

- ✧ Electromagnetic waves⁵⁻¹⁰[*Phys. Rev. Lett.* 114, 127401, (2015); *Nat. Mater.* 15, 542-548, (2016); *Nat. Commun.* 8, 16023, (2017); *Phys. Rev. Lett.* 120, 217401, (2018); *Phys. Rev. Lett.* 123, 103901 (2019)].
- ✧ Airborne sounds^{11,12}[*Nat. Phys.* 12, 1124-1129, (2016); *Phys. Rev. Lett.* 118, 084303, (2017)].
- ✧ Elastic waves¹³⁻¹⁶[*Nat. Commun.* 6, 8682, (2015); *Nature* 564, 229-233, (2018); *Phys. Rev. X* 8, 031074, (2018); *Nat. Commun.* 9, 3072, (2018)].

The *PRApplied* paper reports the realization of QSH for the SAWs. In this phononic crystal, the topological invariant is the spin Chern number C_s , obtained by calculating the integral of Berry curvature over the 1st Brillouin zone. The C_s of ordinary insulators (OIs) is zero, while topological insulators (TIs) are nonzero. At OI-TI interfaces, there are “spin-momentum locked” edge states (*i.e.*, helical edge states).

- Our current paper is about analogous quantum valley Hall (QVH) states. The existence of these states was proposed in electronics in 2007, and be observed in materials, *e.g.*, graphene¹⁷[*Phys. Rev. Lett.* 99, 236809,

(2007)], bilayer graphene¹⁸[*Nano Lett.* **11**, 3453-3459, (2011)] and transition-metal dichalcogenides¹⁹[*Science* **344**, 1489-1492, (2014)]. Most materials have a honeycomb lattice, leading to linear degeneracies (Dirac cones) at their reciprocal space's high symmetry K/K' points. When their space inversion-symmetry is broken (*e.g.*, by A-B sublattice asymmetry), the K/K' degeneracies open, thus forming energy valleys and bulk bandgaps. Unlike the QSH, the integral of the Berry curvature of the two bands besides the valley bandgap over the 1st Brillouin zone is still zero. However, the Berry curvature is localized with equal magnitude and opposite sign at the K/K' points. At the same time, the momentum distance between the two valleys is relatively large. All these make the inter-valley scattering suppressed.

In recent years, QVH also has been developed into classical wave systems. For example:

- ✧ Electromagnetic waves²⁰⁻²⁶ [*Nat. Commun.* **8**, 1304, (2017); *Phys. Rev. B* **96**, 201402, (2017); *Phys. Rev. Lett.* **120**, 063902, (2018); *Nat. Commun.* **10**, 872, (2019); *Nature Nano.* **14**, 31, (2019); *Nat. Photon.* **14**, 446-451, (2020); *Phys. Rev. Lett.* **126**, 230503, (2021);].
- ✧ Airborne sounds²⁷⁻²⁹[*Phys. Rev. Lett.* **116**, 093901, (2016); *Nat. Phys.* **13**, 369-374, (2016); *Nat. Commun.* **11**, 762, (2020)].
- ✧ Elastic waves^{30,31}[*Nat. Mater.* **17**, 993-998, (2018); *Sci Adv* **7**, eabe1398, (2021)].

Our current paper reports the experimental realization of QVH for the SAWs. More importantly, the one-dimensional SAW topological valley-locked edge states (TVSs) are further developed to the quasi-two-dimensional extended topological valley-locked edge states (ETVSs).

Here is an incomplete table listing realizations of these two physical effects in different classical wave systems.

	analogous quantum spin Hall (QSH)	analogous quantum valley Hall (QVH)
		Electromagnetic waves	Phys. Rev. Lett. 114 , 223901, (2015) Nat. Mater. 15 , 542-548, (2016) Nat. Commun. 8 , 16023, (2017) Phys. Rev. Lett. 120 , 217401, (2018) Phys. Rev. Lett. 123 , 103901, (2019)	Phys. Rev. Lett. 120 , 063902, (2018) Nat. Commun. 8 , 1304, (2017); Phys. Rev. B 96 , 201402, (2017) Nat. Commun. 10 , 872, (2019) Nature Nano. 14 , 31, (2019) Nat. Photon. 14 , 446-451, (2020) Phys. Rev. Lett. 126 , 230503, (2021)
Airborne sounds	Nat. Phys. 12 , 1124-1129, (2016); Phys. Rev. Lett. 118 , 084303, (2017)	Nat. Phys. 13 , 369-374, (2016); Nat. Commun. 11 , 762, (2020)
Elastic waves	Nat. Commun. 6 , 8682, (2015); Nat. Commun. 9 , 3072, (2018) Nature 564 , 229-233, (2018); Phys. Rev. X 8 , 031074, (2018);	Nat. Mater. 17 , 993-998, (2018); Sci Adv 7 , eabe1398, (2021)

II. Difference in application prospects and advantages

➤ (★★★) **Matching between the SAW waveguides and transducers.**

In the design of SAW transducers (*i.e.*, the interdigital transducers, IDTs), large-aperture IDTs can guarantee broad working bandwidth and sufficient device power, and are more conducive to impedance matching of traditional 50Ω cables. The *PRApplied* paper does not extend (widen) the SAW waveguide; as a critical improvement, the current paper utilizes a semi-metallic region to extend (widen) the SAW waveguides. These extended (widened) SAW waveguides can (much!) better match the necessary large-aperture IDTs, *i.e.*, a (much!) more proportion of the planer SAWs electrically generated by the IDTs can be injected into the waveguides, as a simple schematic diagram shown in Fig. R3-1.

Figure R3-1 | Planer SAWs, electrically generated by the same IDTs, incident to (a) an un-extended topological valley-locked waveguide and (b) an extended (widened) valley-locked waveguide, respectively. The yellow arrows indicate the SAWs.

From the perspective of the SAW devices, these extended (widened) SAW waveguides that are matched with the IDTs have important advantages. For example, using the same SAW transducers, the *PRApplied* paper requires multiple waveguides to demonstrate a ~15dB waveguide signal in the S_{21} spectrum (see Fig. 4 in the *PRApplied* paper), but in our current paper, only one extended (widened) waveguide is enough! This is undoubtedly a huge contribution to practical topological signal processing using SAWs. For the \$S_{21}\$ signal improvement, see more in our last response to Reviewer Comment #3-5.

➤ (★) **Effective working bandwidth.**

The bandwidth of the SAW waveguides (only un-extended studied) in the *PRApplied* paper is 2.7%. In the current paper, the bandwidth of the un-extended SAW waveguides is 6.6%, and the extended (widened) waveguides are 5.1%(3 layers), 4.8%(5 layers),4.4%(7 layers). In theory, it is much difficult to increase the working bandwidth of the waveguides in the *PRApplied* paper; the material needs to be optimized, and the distance between adjacent micro-pillars will be pushed to the processing limit. As a comparison, for the valley-locked SAW waveguides invented in the current paper, the working bandwidth of the un-extended waveguides can easily exceed 10%, while the extended (widened) waveguides 7.8%. Also, the distance between adjacent pillars has not been reduced, so there is no additional challenge for sample processing.

Here is a brief comparison between these two works

	The PRApplied work	This work
wave system	SAW (quasi Rayleigh modes)	SAW (quasi Rayleigh modes)

physical basis	QSH	QVH & ETVSs
compatibility with SAW transducers	poor	good
doping on the edge & configurability	no	yes
achieved working bandwidth	2.7% extended: not yet studied	un-extended: 6.6% extended: 5.1%, 4.8%, 4.4% (3, 5, 7 layers)
theoretical working bandwidth	un-extended: 10% extended: not yet studied	un-extended: 13% extended: 7.8%, 6.7%, 5.3% (3, 5, 7 layers)

In summary, comparing the *PRApplied* work and our current work, they are firstly different in their physical basis, *i.e.*, analogous QSH v.s QVH; importantly, the former is far superior to the latter in application potentials, main due to the extended topological valley-locked edge states. *As per the reviewer's comment*, the advantage of the current paper over the *PRApplied* paper, especially the matching between the SAW waveguides and transducers, are highlighted more in our revised manuscript and carefully discussed in Note 10 of the SI.

Reviewer Comment #3-3:

How can the C_{6v} crystal in the Physical Review Applied paper cited above show a double Dirac cone at the Gamma point, whereas figure 1 in the present paper shows a single Dirac cone at the K point? As a note, the working frequency is 76 MHz in both cases and the samples look exactly the same.

Our Response: As we replied in **Comment #3-2**, the *PRApplied* paper focuses on QSH states for SAWs, while the current paper implement QVH states (both TVSs and ETVSs) for SAWs. Although the two use the same pillar-type phononic crystals, their topological edge states are essentially different. In the *PRApplied* paper (and many similar papers in other classical wave systems^{5-7,11-16}), the key physics behind realizing the bosonic analogue of QSH is to mimic the spin degrees of freedom so as to create a double Dirac cone, where Kramers doublet exists in the form of pseudospin-up and pseudospin-down. **These 4-fold Dirac cones appearing at the Gamma point are generally constructed using the so-called “zone folding” method^{6,12}[*Phys. Rev. Lett.* 114, 223901 (2015); *Phys. Rev. Lett.* 118, 217401 (2017)], which requires a C_{6v} hexagonal lattice with two 2D irreducible group representations at Gamma point. However, in the current paper of valley cases, 2-fold Dirac cones at the K/K' points formed by standard honeycomb lattice are sufficient (only one 2D representation at K/K' point). After opening the valley bandgap, the C_{6v} symmetry is reduced to C_{3v} without 2D representation at the K/K' points.**

At the same time, we need to explain that the 70-80MHz SAW phononic crystal is a relatively mature experimental system for us, with a stable sample preparation process and complete testing capabilities. We have published a series of works on this system. Of course, we can also extend this type of SAW phononic crystal to other frequencies.

Moreover, the complete band structure must be shown, not only the small frequency range around 76 MHz.

Action Taken: As per the reviewer's comment, the complete band structures are shown in Fig. R3-2, also Fig. S7 of the SI.

Figure R3-2 | a, b, c: Band structures of full frequencies for our SAW valley insulator A (VA), semi-metal (SM), and valley insulator B (VB), respectively.

Reviewer Comment #3-4:

Is the anisotropy of LNO taken into account in the design and the numerical simulations?

Our Response: In the simulation, we have considered the anisotropy and piezoelectricity of the LNO.

Does the C_{6v} symmetry of the 2D phononic crystal survive the trigonal crystal system of LNO?

Our Response: We appreciate the reviewer's comment on the very important point around the symmetry of the 2D phononic crystal demonstrated in this paper. As the reviewer said, the trigonal crystal of LNO substrate slightly affects our phononic crystal's symmetry, making it not strictly C_{6v} . However, this effect is faint and does not affect the topological properties of the phononic crystal.

Below we give some explanations on this issue:

First, to show the influence of the anisotropy of LNO on the 2D phononic crystal, we calculated the phononic crystal's band structures and equal frequency contours (EFCs) with and without considering the anisotropy. In the isotropy case, Young's modulus, density, and Poisson's ratio of a hypothetical LNO material are set to 203GPa, 4700 kg m⁻³, and 0.31, respectively, and the results are shown in the upper panel of Fig. R3-3. As a comparison, the results of a real LNO case (both anisotropy and piezoelectricity are taken into account) are shown in the lower panel of the same figure.

Figure R3-3 | a-c: hypothetical isotropic LNO. d-f: real LNO with anisotropy and piezoelectricity. a/d: band structures of the 2D phononic crystal along Γ -K-M-K'- Γ and Γ - \bar{K} -M- \bar{K}' - Γ . b/e: 3D band structure of the 1st BZ. c/f: equal frequency contours (EFCs) of the 1st BZ.

In the hypothetical isotropic LNO, the SAW Dirac points of the phononic crystal are strictly located at the high-symmetric K and K' points of the C_{6v} lattice. In real LNO with anisotropy and piezoelectricity, the SAW Dirac points still exist, and the band structure has not changed much. Still, affected by the anisotropy LNO substrate, the SAW Dirac points will be slightly offset along the z-direction in the EFC diagram. We use \bar{K} and \bar{K}' to

indicate the positions of SAW Dirac points at this time. The distance between \bar{K} and K is only about 1/100 of the distance between Γ and K.

Second, to demonstrate that the anisotropy of LNO will not affect the topological properties of the 2D phononic crystal, we further calculated the Berry curvature of the isotropic LNO and the real LNO, as shown in Fig. R3-4. In both cases, the Berry curvature has a local distribution of equal magnitude and opposite sign at the K(\bar{K}) and K'(\bar{K}') points, thus supporting the QVH states.

Figure R3-4 | **a**: Berry curvature around K and K' points in the hypothetical isotropic LNO case. **b**: Berry curvature around \bar{K} and \bar{K}' points in the real LNO case with anisotropy and piezoelectricity.

Third, we conduct some analysis on the formation of valley-locked edge states under the influence of LNO anisotropy, as shown in Fig. R3-5. R3-5a shows the Zigzag interface formed by valley insulator A (VA) and valley insulator B (VB) in the real space. R3-5b and R3-5c show the Berry curvature at the VA-VB interface in the isotropic LNO case and the real LNO case, respectively. The distributions of Berry curvature at K (\bar{K}) or K' (\bar{K}') inverted when crossing the Zigzag interface, leading to nonzero valley-projected Chern number. Thus, valley-locked topological edge states will certainly appear along with the Zigzag interface under both circumstances, according to bulk-edge correspondence³². There are discussions about the effect of material symmetry on the symmetry of phononic crystals³³.

Figure R3-5 | **a:** Zigzag interface formed by valley insulator A (VA) and valley insulator B (VB) in real space. **b:** Berry curvature distribution in the isotropy LNO on both sides of the interface. **c:** Berry curvature distribution in anisotropy LNO on both sides of the interface. The offset of the \bar{K} (\bar{K}') point to the original K (K') point is magnified by 10 times in the figure to show their difference clearly.

Our Response & Action Taken: *According to the reviewer's question*, we have added a related description on this symmetry issue in the main text of our revised manuscript, as "... the symmetry of y-cut LiNbO₃ substrate slightly affects our phononic crystal's symmetry. However, this effect is faint and does not affect the presence of TVSS" in the 2nd paragraph of Section I and double-checked all the Dirac frequencies in our manuscript. Also, we put the response to this question (including the figures) as Note 7 in SI.

Reviewer Comment #3-5:

It is argued that the topological waveguides created are broadband. This statement is particularly misleading. The fractional bandwidth varies between between 1/15 and 1/25 (3 to 5 MHz around 75 MHz); I would classify such values as small bandwidth and this is a general problem for topological waveguides designed from the opening of the band gap that was originally closed by symmetry.

Our Response: We thank the reviewer's comment about the description of the bandwidth of our SAW waveguides. We fully agree with the reviewer that increasing the working bandwidth is an important challenge in the current research on electromagnetic/mechanical topological materials. We are sorry for the misleadingness caused by our English expression's inaccuracy. In our original manuscript, we tried to use the description of "broadband" as an opposing concept to "narrowband". However, many similar descriptions such as "broadband", "wideband", "ultra-broadband" and "ultra-wideband" do confuse people (including us). We appreciate the reviewer's understanding of this issue.

Action Taken: To avoid unnecessary misunderstandings, we have removed similar descriptions in the revised manuscript and replaced them with "considerable working bandwidth" or accurate bandwidth value. Also, we have added a description of the bandwidth problem in the "Conclusion & Discussions" section of our revised manuscript, including some related references^{34,35}.

"One challenge is how to increase the working bandwidth of these ETVs further. If the bandwidth can exceed 15% or even 25% (*i.e.*, the largest bandwidth in current 5G mobile networks), it will significantly facilitate their demonstrating functionalities in practical scenarios. Some valuable results have recently appeared in broadband topology materials, *e.g.*, those using topology optimizations³⁴ and inverse design³⁵."

Reviewer Comment #3-5:

The figures need to be prepared with more care.

Our Response: We are grateful to the reviewer for the careful attention and helpful suggestions on the figures of our manuscript. According to them, we have redrawn many of them in the revised manuscript.

The wavevector axis of phononic band structures shows inconsistent information.

Our Response & Action Taken: We are confused about this comment because we have not found inconsistent information in the wavevector axis of phononic band structures, but we try to respond to it and redraw Figs. 1d-1f in a more clear fashion.

The band structures we have calculated and displayed are divided into two categories in this paper.

- One is the band structures of 2D phononic crystals, as shown in Fig. 1. These band structures are drawn according to the path of several high symmetry points in the 1st Brillouin zone of the 2D reciprocal space, *i.e.*, from point Γ to K, to M, and back to Γ .
- The other is the band structures of 1D topological edges (*i.e.*, waveguides), as shown in Fig. 2. These band structures need to be drawn using supercells^{36,37}, and are illustrated in the whole 1st Brillouin zone of the 1D reciprocal space, *i.e.*, from $-\pi/a$ to π/a .

The SEM images are heavily processed (look like CAD images) and no details of the actual fabricated samples are provided.

Our Response & Action Taken: We thank the reviewer for his/her helpful comments on SEM images and fabrication details of our samples. Due to our need to draw 3D graphics in Figs. 2 to 5, the original 2D SEM images have been distorted. To accurately display the 3D information of our samples, we replaced all the 2D SEM images in these figures with 3D CAD images, and placed the SEM images as a new part in the SI. Also, we have newly drawn a detailed figure of sample processing and put it together with the SEM images.

Figure S1 | a: spin coating of photoresist (S1813, $\sim 1\mu\text{m}$). b: UV lithography. c: seed layer deposition.

e: photoresist removing (S1813). f: spin coating of photoresist (AZ9260, ~10 μ m). g: UV lithography.
h: electroplating for nickel (Ni) micropillars. i: photoresist removing (AZ9260).

SEM images of the fabricated SAW PnC samples
(all scale bars represent 50 μ m)

Figure S2 | Samples in Figs. 2d, 2e, and 2f

Figure S3 | Samples in Fig. 3

Figure S4 | Samples in Figs. 4a and 4b

Figure S5 | Samples in Figs. 5a, 5b, and 5c

Figure S6 | Samples in Fig. 8c

Color bars have no units or scale.

Our Response & Action Taken: We thank the careful attention of the reviewer for the unit and scale of the color bars. In some color bars that display intensity (e.g., SAW amplitude, energy), for the concise of the information, we normalized our experimental data, as a common way in many references^{13,21,29} [*Nat. Commun.* **6**, 8682, (2015); *Nat. Commun.* **10**, 872, (2019); *Nat. Commun.* **11**, 762, (2020)]. **As per the reviewer's comment**, we replaced the “*max*” and “*min*” with scales on the original color bars. Also, in the figure captions, we clarified the physical meaning of these color bars, e.g., “**out-of-plane displacement**” and “**out-of-plane amplitude**”.

The 4-port splitter in figure 3 is a 3-port splitter per design.

Our Response & Action Taken: We appreciate the reviewer for carefully pointing out the wrong description related to Fig.4 in our paper. **As per the reviewer's comment**, we have changed the related description from “a four-ports SAW splitter” to “a three-ports SAW splitter”.

The transmission values in figure S4 show strong insertion loss, from 11 dB to 26 dB; the conclusion that 'SAWs transmission [...] has unprecedented high throughput' has no ground and no comparison with previous results in the literature is presented.

Our Response: We appreciate the reviewer's comment on the important point around the SAW transmission demonstrated in this paper. This is related to one of the most critical contributions of our work, which we will explain below.

First, we want to clarify that “high throughput” is a description relative to common designs. Compared with common topological waveguides that have not been extended (widened), an extended topological waveguide inherit the “anti-reflection” ability (making the waveguide have a high degree of freedom and low loss), and at the same time, its flow unprecedentedly increases several times or even dozens of times. It is similar to comparing a small stream and a big river. Therefore, we call it “high throughput”. Perhaps this description is easy to confuse, according to the reviewer’s comment, we have changed it to “high-flow” in our revised manuscript.

Second, the reviewer mentioned the insertion loss in Fig. S4 of our original manuscript. This figure is used to illustrate the advantages of ETVSSs, i.e., reducing the loss from 26dB to 11dB. In this figure, all these insertion losses (from 11dB to 26dB) are measured from electrical S_{21} spectra between our SAW generating and receiving transducers. In fact, these losses mainly come from the mismatch between the SAW generating transducer and the SAW waveguide, i.e., the SAWs electrically excited by the transducer cannot ALL enter the waveguide. A simple schematic diagram is shown in Fig. R3-6.

Figure R3-6 | **a** and **b**: planer SAWs, electrically generated by the same transducer, incident to (a) an un-extended topological valley-locked waveguide and (b) an extended (widened) valley-locked waveguide, respectively. The yellow arrows indicate the SAWs. **c** and **d**: electrically measured S_{21} spectra between the SAW generating and receiving transducers for **a** and **b**, respectively. **Signal brought by the SAW waveguide has a ~15dB improvement in the latter.**

As we all know, in the design of SAW transducers (i.e., the interdigital transducers, IDTs), large-aperture IDTs

can guarantee broad working bandwidth and sufficient device power, and are more conducive to impedance matching of traditional 50Ω cables. In our experiments, because we need to perform on-chip testing from 65MHz to 85MHz (about 30% bandwidth), the aperture of our IDTs is very large (2.5mm, about 50-60 times the SAW wavelength). Consequently, only a part of the SAW electrically excited by the IDT can enter the waveguide for transmission, and finally be received by another IDT on the emitting end.—This whole process brought about the insertion loss (from 11dB to 26dB) that the reviewer mentioned.

R3-6a shows the situation when the topological waveguide is not extended (widened). It is exactly the “ground” mentioned by the reviewer, acting as a control sample in our experiments. In this case, it is obvious that only a small portion of the SAWs excited by IDTs can enter the waveguide, so the insertion loss is large (~26dB), and the resolution of the signal transmitted by the waveguide is poor (the blue area in the figure).

As a comparison, R3-6b shows the situation after the topological waveguide has been extended (widened). At this time, a greater proportion of SAWs can enter the waveguide, thereby considerably reducing the insertion loss. For waveguides with 3, 5, and 7 doping layers, the insertion loss is reduced by about 9dB (from 26dB to 17dB), 15dB (from 26dB to 11dB), and 14dB (from 26dB to 12dB), respectively. Although the current experimental data cannot show a quantitative relationship, it is enough to demonstrate the advantages of these extended topological waveguides qualitatively. They greatly match the topological waveguides and practical transducers for SAWs, thereby improving the resolution of the signal transmitted by the waveguides.

Figure R3-7 | **a** and **b**: planer SAWs, generated by transducers with **(a)** 2mm aperture and **(b)** 1.2mm aperture, incident to the same extended (widened) waveguide. **c**: simulated S_{21} spectra of the two different pairs of SAW generating and receiving transducers.

When the aperture of the SAW waveguide is closer to the aperture of the transducer, this insertion loss will be further reduced. For example, we simulated the situation when the transducer aperture was reduced from 2mm (~50 times of wavelength) to 1.2mm (~30 times of wavelength), as shown in Fig. R3-7. For a waveguide with a certain width, when the transducer aperture is reduced to 60% of the previous one, in the S_{21} spectrum between

the generating and receiving transducers, the signal brought by the waveguide is improved by about 3dB.

Action Taken: According to the reviewer's comment, we first changed the “high throughput” to “high-flow” to avoid misunderstanding. Also, we have added a detailed description of the matching of SAW waveguides and SAW transducers (including Figs. R3-6 and R3-7) in the SI.

Our General Response: In the end, we once again thank Reviewer #3 for his/her careful review and valuable comments on our work. These comments help us to improve the paper to a better level. We sincerely hope that Reviewer #3 now agrees with us and Reviewers #1 and #2 that our work has enough novelty and impact to merit a publication in Nature Communications.

- 1 Murakami, S., Nagaosa, N. & Zhang, S.-C. Dissipationless quantum spin current at room temperature. *Science* **301**, 1348-1351, (2003).
- 2 Sinova, J. *et al.* Universal Intrinsic Spin Hall Effect. *Phys. Rev. Lett.* **92**, 126603, (2004).
- 3 Kane, C. L. & Mele, E. J. Quantum Spin Hall Effect in Graphene. *Phys. Rev. Lett.* **95**, 226801, (2005).
- 4 Bernevig, B. A., Hughes, T. L. & Zhang, S. C. Quantum spin Hall effect and topological phase transition in HgTe quantum wells. *Science* **314**, 1757-1761, (2006).
- 5 Ma, T., Khanikaev, A. B., Mousavi, S. H. & Shvets, G. Guiding electromagnetic waves around sharp corners: topologically protected photonic transport in metawaveguides. *Phys. Rev. Lett.* **114**, 127401, (2015).
- 6 Wu, L. H. & Hu, X. Scheme for Achieving a Topological Photonic Crystal by Using Dielectric Material. *Phys. Rev. Lett.* **114**, 223901, (2015).
- 7 Yves, S. *et al.* Crystalline metamaterials for topological properties at subwavelength scales. *Nat. Commun.* **8**, 16023, (2017).
- 8 Cheng, X. *et al.* Robust reconfigurable electromagnetic pathways within a photonic topological insulator. *Nat. Mater.* **15**, 542-548, (2016).
- 9 Yang, Y. *et al.* Visualization of a Unidirectional Electromagnetic Waveguide Using Topological Photonic Crystals Made of Dielectric Materials. *Phys. Rev. Lett.* **120**, 217401, (2018).
- 10 Smirnova, D. *et al.* Third-Harmonic Generation in Photonic Topological Metasurfaces. *Phys. Rev. Lett.* **123**, 103901, (2019).
- 11 He, C. *et al.* Acoustic topological insulator and robust one-way sound transport. *Nat. Phys.* **12**, 1124-1129, (2016).
- 12 Zhang, Z. *et al.* Topological Creation of Acoustic Pseudospin Multipoles in a Flow-Free Symmetry-Broken Metamaterial Lattice. *Phys. Rev. Lett.* **118**, 084303, (2017).
- 13 Mousavi, S. H., Khanikaev, A. B. & Wang, Z. Topologically protected elastic waves in phononic metamaterials. *Nat. Commun.* **6**, 8682, (2015).
- 14 Cha, J., Kim, K. W. & Daraio, C. Experimental realization of on-chip topological nanoelectromechanical metamaterials. *Nature* **564**, 229-233, (2018).
- 15 Miniaci, M., Pal, R. K., Morvan, B. & Ruzzene, M. Experimental Observation of Topologically Protected Helical Edge Modes in Patterned Elastic Plates. *Phys. Rev. X* **8**, 031074, (2018).
- 16 Yu, S. Y. *et al.* Elastic pseudospin transport for integratable topological phononic circuits. *Nat. Commun.* **9**, 3072, (2018).
- 17 Xiao, D., Yao, W. & Niu, Q. Valley-contrasting physics in graphene: magnetic moment and topological transport. *Phys. Rev. Lett.* **99**, 236809, (2007).

- 18 Qiao, Z., Jung, J., Niu, Q. & Macdonald, A. H. Electronic highways in bilayer graphene. *Nano Lett.* **11**, 3453-3459, (2011).
- 19 Mak, K. F., McGill, K. L., Park, J. & McEuen, P. L. The valley Hall effect in MoS₂ transistors. *Science* **344**, 1489-1492, (2014).
- 20 Noh, J., Huang, S., Chen, K. P. & Rechtsman, M. C. Observation of Photonic Topological Valley Hall Edge States. *Phys. Rev. Lett.* **120**, 063902, (2018).
- 21 He, X.-T. *et al.* A silicon-on-insulator slab for topological valley transport. *Nat. Commun.* **10**, 872, (2019).
- 22 Chen, Y. *et al.* Topologically Protected Valley-Dependent Quantum Photonic Circuits. *Phys. Rev. Lett.* **126**, 230503, (2021).
- 23 Wu, X. *et al.* Direct observation of valley-polarized topological edge states in designer surface plasmon crystals. *Nat. Commun.* **8**, 1304, (2017).
- 24 Gao, Z. *et al.* Valley surface-wave photonic crystal and its bulk/edge transport. *Phys. Rev. B* **96**, 201402, (2017).
- 25 Shalaev, M. I., Walasik, W., Tsukernik, A., Xu, Y. & Litchinitser, N. M. Robust topologically protected transport in photonic crystals at telecommunication wavelengths. *Nature Nanotechnology* **14**, 31-34, (2019).
- 26 Yang, Y. *et al.* Terahertz topological photonics for on-chip communication. *Nat. Photon.* **14**, 446-451, (2020).
- 27 Lu, J., Qiu, C., Ke, M. & Liu, Z. Valley Vortex States in Sonic Crystals. *Phys. Rev. Lett.* **116**, 093901, (2016).
- 28 Lu, J. *et al.* Observation of topological valley transport of sound in sonic crystals. *Nat. Phys.* **13**, 369-374, (2016).
- 29 Tian, Z. *et al.* Dispersion tuning and route reconfiguration of acoustic waves in valley topological phononic crystals. *Nat. Commun.* **11**, 762, (2020).
- 30 Yan, M. *et al.* On-chip valley topological materials for elastic wave manipulation. *Nat. Mater.* **17**, 993-998, (2018).
- 31 Xi, X., Ma, J., Wan, S., Dong, C. H. & Sun, X. Observation of chiral edge states in gapped nanomechanical graphene. *Sci Adv* **7**, eabe1398, (2021).
- 32 Rudner, M. S., Lindner, N. H., Berg, E. & Levin, M. Anomalous Edge States and the Bulk-Edge Correspondence for Periodically Driven Two-Dimensional Systems. *Phys. Rev. X* **3**, 031005, (2013).
- 33 Li, S., Kim, I., Iwamoto, S., Zang, J. & Yang, J. Valley anisotropy in elastic metamaterials. *Phys. Rev. B* **100**, 195102, (2019).
- 34 Christiansen, R. E., Wang, F. & Sigmund, O. Topological Insulators by Topology Optimization. *Phys. Rev. Lett.* **122**, 234502, (2019).
- 35 Nussbaum, E., Sauer, E. & Hughes, S. Inverse design of broadband and lossless topological photonic crystal waveguide modes. *Opt. Lett.* **46**, 1732-1735, (2021).
- 36 Lin, L.-L. & Li, Z.-Y. Interface states in photonic crystal heterostructures. *Phys. Rev. B* **63**, 033310, (2001).
- 37 Ao, X., Lin, Z. & Chan, C. T. One-way edge mode in a magneto-optical honeycomb photonic crystal. *Phys. Rev. B* **80**, 033105, (2009).

REVIEWERS' COMMENTS

Reviewer #1 (Remarks to the Author):

In the revised version of the manuscript, the quality of Fig 2d has been improved greatly. Now it is my pleasure to recommend this manuscript for publication in Nature Communications.

Reviewer #2 (Remarks to the Author):

The authors have carefully addressed my comments in their responses. In the current form, I can suggest the publication of this paper on Nature Communications.

Reviewer #3 (Remarks to the Author):

I have been globally convinced by the answers provided by the authors and by the revision they performed. I believe the paper, the methodology and the novelty are now much clearer.

Regarding the concurrent submission of the paper to Phys. Rev. Applied, I agree that the topological invariants used are different, but I am still surprised that the less interesting solution was submitted after the present one to Nature Commun. This remark, however, does not affect the contents of the present manuscript.

I appreciate the efforts made by the authors to describe the improvements their topological waveguide provides. The result is still far from industrial applications, in my opinion, but this on-going research is important and must be performed.

The results obtained are noteworthy and the conclusions are supported.

Overall I have no further comments.

Author Response to Reviewer Comments

Reviewer #1 (Remarks to the Author):

In the revised version of the manuscript, the quality of Fig 2d has been improved greatly. Now it is my pleasure to recommend this manuscript for publication in Nature Communications.

Our Response: We are grateful to the reviewer for his/her reviewing and recommendation of our work.

Reviewer #2 (Remarks to the Author):

The authors have carefully addressed my comments in their responses. In the current form, I can suggest the publication of this paper on Nature Communications.

Our Response: We once again thank the reviewer for his/her reviewing and recommendation of our work.

Reviewer #3 (Remarks to the Author):

I have been globally convinced by the answers provided by the authors and by the revision they performed. I believe the paper, the methodology and the novelty are now much clearer.

Our Response: We are very pleased and appreciate the reviewer's recognition of our revision, the methodology and the novelty of our work.

Regarding the concurrent submission of the paper to Phys. Rev. Applied, I agree that the topological invariants used are different, but I am still surprised that the less interesting solution was submitted after the present one to Nature Commun. This remark, however, does not affect the contents of the present manuscript.

Our Response: In this regard, we thank the reviewer's understanding and supporting.

I appreciate the efforts made by the authors to describe the improvements their topological waveguide provides. The result is still far from industrial applications, in my opinion, but this on-going research is important and must be performed.

Our Response: We thank the reviewer's recognition of our topological waveguide and for pointing out the importance of this on-going research. On the basis of current paper, we will make continuous efforts towards practical industrial applications, as the reviewer mentioned.

The results obtained are noteworthy and the conclusions are supported.

Overall I have no further comments.

Our Response: We once again thank the reviewer for his/her careful review. His/her comments and suggestions have greatly helped us to improve the quality of the present paper.